# Digital health literacy, online information-seeking behaviour, and satisfaction of Covid-19 information among the university students of East and South-East Asia

Mila Nu Nu Htay[1], Laurence Lloyd Parial[2,3], Ma. Carmen Tolabing[4], Kevin Dadaczynski[5,6], Orkan Okan[7], Angela Yee Man Leung[2‡], Tin Tin Su[8‡]*

1 Department of Community Medicine, Faculty of Medicine, Manipal University College Malaysia, Bukit Baru, Melaka, Malaysia, 2 School of Nursing, The Hong Kong Polytechnic University, Hong Kong SAR, China, 3 College of Nursing, University of Santo Tomas, Manila, Philippines, 4 Department of Epidemiology and Biostatistics, College of Public Health, University of the Philippines, Manila, Philippines, 5 Center for Applied Health Science, Leuphana University Lueneburg, Lueneburg, Germany, 6 Department of Health Science, Fulda University of Applied Sciences, Fulda, Germany, 7 Department of Sport and Health Sciences, Technical University Munich, München, Germany, 8 South East Asia Community Observatory (SEACO) & Global Public Health, Jeffery Cheah School of Medicine and Health Sciences, Monash University Malaysia, Bandar Sunway, Malaysia

‡ AYML and TTS are joint senior authors on this work.
* TinTin.Su@monash.edu

## Abstract

During the COVID-19 pandemic, there is a growing interest in online information about coronavirus worldwide. This study aimed to investigate the digital health literacy (DHL) level, information-seeking behaviour, and satisfaction of information on COVID-19 among East and South-East Asia university students. This cross-sectional web-based study was conducted between April to June 2020 by recruiting students from universities in China, Malaysia, and the Philippines. University students who have Internet access were invited to participate in the study. Items on sociodemographic variables, DHL, information-seeking behaviour, and information satisfaction were included in the questionnaire. Descriptive statistics and logistic regression analysis were conducted. A total of 5302 university students responded to the survey. The overall mean score across the four DHL subscales was 2.89 (SD: 0.42). Search engines (e.g., Google, Bing, Yahoo) (92.0%) and social media (88.4%) were highly utilized by the students, whereas Websites of doctors or health insurance companies were of lower utilization (64.7%). Across the domains (i.e., adding self-generated content, determining relevance, evaluating reliability, and protecting privacy) higher DHL was positively associated with higher usage of trustworthy resources. Providing online information on COVID-19 at official university websites and conducting health talks or web-based information dissemination about the strategies for mental health challenges during pandemic could be beneficial to the students. Strengthening DHL among university students will enhance their critical thinking and evaluation of online resources, which could direct them to the quality and trustworthy information sources on COVID-19.

**Data Availability Statement:** The minimum underlying data set is in the supporting information files.

**Funding:** The author(s) received no specific funding for this work.

**Competing interests:** The authors have declared that no competing interests exist.

# Introduction

The emergence of the novel coronavirus (COVID-19) is a major global health challenge due to the rapid transmission and disease uncertainty, particularly at the beginning of the pandemic [1, 2]. In addition to the physical health impacts of COVID-19, there is a rapid amplification of both accurate and inaccurate information on digital platforms worldwide [3]. The improvement in digital technology and internet access increased online searching for health-related information among the public [4]. A study conducted in the United States of America analysing tweets from 3 million people reported that false news spreads faster in social media such as Twitter than factual information [5]. Similarly, higher engagement of false information was reported on Facebook just before the US presidential campaign [6]. The broad public reach of the Internet and social media favours the spread of rumours and dissemination of unreliable information [7]. The overabundance of information, or the *infodemic* (short of 'information epidemic'), is threatening to people, making it more difficult to search for verified facts about COVID-19. The World Health Organization suggested combating infodemic by reporting the misinformation to the authorities, disseminating the accurate information, and implementing action plans in respective countries [8]. In this context, health literacy is an important concept [9].

Health literacy is the ability to find, understand, appraise, and apply health information in order to make decisions about health and wellbeing across the domains of health care, disease prevention, and health promotion [10]. People's ability to search the information from electronic sources, critical review, and use this information for their health-related decision-making is known as digital health literacy (DHL) or eHealth literacy [4, 11]. Health literacy is an important concept during the COVID-19 pandemic, when health information is available mostly in digital and online forms [12], while people need the skills to navigate these information environments and process and filter information to make health decisions. DHL varies among people depending on their literacy levels and socioeconomic status, presenting health issues, motivation to seek online information, and familiarity with technology and electronic gadgets [11, 13]. A recent study on coronavirus-specific health literacy conducted in Germany in Spring 2020 reported that half of the adult population (50.1%) had problematic or inadequate health literacy [14]. They felt less-informed about the COVID-19 and confused by the amount of information. Moreover, 47.8% of adults were found to experience difficulties judging the trustworthiness of information from (online) media [14].

Another study conducted in five Western countries, including the U.K., Ireland, USA, Spain, and Mexico, reported that exposure to misinformation negatively influenced the people's compliance with COVID-19 prevention guidelines and their willingness to receive vaccination for themselves and recommend it for other vulnerable people [15]. In Italy, misinformation and racism-related information were still reported in some regions [16]. During this pandemic, frequent social media exposure was significantly associated with anxiety and depression [17].

During the COVID 19 pandemic, university student population is facing psychological impact threatening their health and well-being and at the same time they experience challenges when seeking reliable online information [18]. A study among students in Australia revealed that approximately two-thirds of the students have low or very low self-perceived psychological well-being [18]. Dadaczynski et al. reported difficulties in finding the correct information on the specific health-related topics, evaluating the reliability of online information among university students in Germany [19]. Similarly, students in Slovenia faced difficulties in accessing the reliability of the online information [20]. Therefore, it is vital to understand the information-seeking behaviours and digital health literacy to formulate appropriate risk communication strategies for COVID-19. However, there is scarce information about these aspects among young adults in Asia. Previous studies in Southeast Asia reported a high prevalence of limited

health literacy in Malaysia (94.2% among caregivers in 2015 and 85.8% among type 2 DM patients in 2016), while in the neighbour countries such as Singapore (43.7%), Thailand (48.7%) and Myanmar (28.2%) a lower percentage could be found among patients, caregivers, and general population [21–25]. A study conducted among middle school students in the Philippines before the COVID-19 pandemic reported limited health literacy especially on personal health and on prevention and control of diseases [26]. A meta-analysis of health literacy among Chinese students revealed that only 26% had an adequate health literacy level [27]. The wealth of information provided during the COVID-19 pandemic caused confusion and negative impacted individual prevention and control measures especially in limited health literate population groups. Therefore, there is a need to investigate DHL and information seeking pattern during the pandemic period in Asian countries.

With improving access to digital devices and the internet, individuals usually seek health information online. A study conducted in Greece with a national representative sample of 1277 adults demonstrated that 90% of the university students used the internet to seek health information and 85.4% were satisfied with the adequacy of the information on health issues by searching on websites, 61.2% on blogs and 34.2% on social networking sites [28]. In Turkey, the adolescent health literacy level was positively associated with their health promoting behaviours and health responsibility [29]. Furthermore, health literacy was found to be positively linked with health awareness and satisfaction among 1055 Turkish adult patients at primary healthcare [30].

In Malaysia and the Philippines, the first wave of COVID-19 started in the first quarter of 2020 [31, 32]. Based on an increase COVID-19 incidence, the respective government-initiated disease control strategies including the movement control order, travel restrictions, prohibition of mass gathering and enforcement on wearing masks [31, 32]. At universities, the teaching-learning mode has been transformed to an online mode. During the early phase of COVID-19 pandemic, knowledge on COVID-19 among the Filipino students was average (63.83%) to good (23.08%) [32]. In China, 56% of college students reported the adequate knowledge about the COVID-19 disease and respective symptoms [33]. In Malaysia, a high level of adequate knowledge could be found for 80.5%, however, the actual behavioural practice of wearing facemask was less common (51.2%) during the early phase of pandemic [34]. Among Filipino university students, social media was one of the main information sources [32]. Some proportion of students in Malaysia also reported about misconception regarding COVID-19 transmission, not following the social distancing, and not practicing hand hygiene [31]. Therefore, it is essential to disseminate timely disease related information and standard operating procedures (SOP) to the population of university students. In Germany, a study of adults reported that lower health literacy level was associated with confusion on COVID-19 related information [14]. Among the people with chronic diseases in mainland China, Hong Kong and Macau, the online information satisfaction during COVID-19 was related to the DHL, online searching of COVID-19 symptoms, their perceived importance of quickly learning from the information search and less frequent searches about dealing with psychological stress [35]. Little is known about the information satisfaction on COVID-19 online information and its associated factors for university students from East and South-East Asia.

Therefore, our study aimed to investigate the digital health literacy, information-seeking behaviours, and satisfaction with COVID-19 information search among university students in East and South-East Asia (China, Malaysia, and the Philippines). The following research questions were pursued in this study

i. What is the level of DHL among university students in China, Malaysia, and the Philippines, including the analysis of their most frequently used online health information sources and topics with regard to COVID-19?

ii. Are there any associations between the DHL level and utilization of trustworthy online sources?

iii. What are the predictors of information satisfaction related to COVID-19 information search?

We assume that higher DHL would be associated with utilization of trustworthy online information sources and hypothesize that higher DHL would be associated with higher utilization of trustworthy sources (such as news portals, health portals, websites of public bodies, doctors, and health insurance). In addition, the study aims at identifying factors associated with higher satisfaction with online health information seeking.

## Methods

### Study design and participants

This study is part of an international health literacy research network (the COVID-19 Health Literacy Global Research Network: 'COVID-HL', https://covid-hl.orghttps://covid-hl.eu/), which has been conducted in more than 40 countries. The web-based cross-sectional study was conducted between 12 April to 09 June 2020 (three to four weeks for each country) among university students in China, Malaysia, and the Philippines. Ethical approval was granted from the Hong Kong Polytechnic University (Ref. No: HSEARS20200407001), Monash University (Project No: 24295), Melaka Manipal Medical College (Ref. No: 5/2020), and University of Santo Tomas–College of Nursing (Ref. No.: USTCON-2020-OR16).

University students were recruited from eighty different universities and with access to the Internet. To disseminate the survey, we utilized various sources such as sending invitation emails through educational institutions, social networking sites (e.g., Facebook, Twitter, Weibo). In the Special Administrative Regions of China (Hong Kong and Macau), Malaysia, and the Philippines, Google Forms was used for the data collection. WenJenXin (WJX) was employed for the data collection in Mainland China. Simplified/traditional Chinese and English versions were used for China. For Malaysia and the Philippines, the English version was utilized. Some items were modified to match the local context in the different study settings (e.g., using local websites as information sources and cities in sociodemographic information). We adopted the World Health Organization process of translating and adapting instruments to convert the original English version to the Chinese language. The forward translation and back-translation method was used to prepare the Chinese version of the questionnaire [36].

Before taking part in the online survey, informed consent was obtained from the university students by requiring them to press the consent button on the first page of the survey link, which contained information on the background and aims of the study, data protection and use as well as ethical information of this study. Participation was voluntary. University students who were willing to complete the survey indicated their consent before accessing the questionnaire. Respondents also had the liberty to skip any question or stop answering at any time. There were a few minors less than 18 years of age who completed the survey. Thus, we removed the responses from minors (n = 49) from the data analysis.

### Measures

The study utilized the COVID-HL University Students Questionnaire, which was developed by the COVID-Health Literacy Consortium [19] and then translated and culturally adapted. The questionnaire is composed of several sections pertaining to sociodemographic information, life situation and future anxiety, health literacy and information seeking behavior, and

personal health situation. Following the objectives of the present paper, we focus on participants' sociodemographic variables, online information-seeking behaviour, DHL, and satisfaction with the information related to COVID-19.

i.  Sociodemographic variables
    Sociodemographic variables included gender (male, female, diverse), country (China, Malaysia, The Philippines), study programme (Health-sciences including Medicine/Biomedical sciences, Allied Health Sciences: Nursing, Pharmacy, Medical Technology; and Non-health sciences including Engineering Sciences/ICT, Linguistics & cultural studies/communication arts, Mathematics/natural sciences, Law and economics/criminology/ public administration/political science, Social sciences/Social work/Psychology/Education, Business/Commerce/Accountancy/Management, Tourism/Hospitality/Hotel management, Architecture/design/arts/visual studies, Secondary level/Others), current study level (undergraduate, post-graduate), and financial source for the study (support by parents, student grant, employment during the semester, employment during the semester break, scholarship, and other). Age was measured in years and categorized as (18–25 years, 26–35 years, ≥36 years).

ii. Digital health literacy
    The Digital Health Literacy Instrument (DHLI) developed by Van der Vaart (2017) was adapted to COVID-19 pandemic as main focus of the COVID-HL Network [37]. While the original DHLI is comprised of 7 subscales, we used the following five domains of (1) information searching or using appropriate strategies to look for information, (2) adding self-generated content to online-based platforms (3) evaluating reliability of online information, (4) determining relevance of online information, and (5) protecting and respecting privacy in Internet use. A total of 15 items (three per each dimension) were asked to the university students, and their answers were recorded with a four-point Likert scale (very difficult = 1, difficult = 2, easy = 3, very easy = 4). The privacy protection subscale answers were recorded as 'often' = 1, 'several times' = 2, 'once' = 3, and 'never' = 4.

iii. Information-seeking behaviour
     University students' information-seeking behaviours were assessed by asking them about the sources of COVID-related information (e.g., search engines, websites) they have accessed. Respondents were presented a list of ten web-based sources that could be rated on a four-point scale (often, sometimes, rarely, never) [19]. The utilization was classified as high utilization (often, sometimes) and low utilization (rarely, never). Moreover, commonly searched topics related to the COVID-19 pandemic were also assessed using a self-developed list of ten topics (e.g., individual measures to protect against infection), which could be answered by 'yes' or 'no' [19]. University students' perceptions of the importance of several factors of online health information search was assessed by six items developed by Gebel et al. (2014): (1) the information is up-to-date, (2) the information is verified, (3) you quickly learn the most important things, (4) the information comes from official sources, (5) different opinions are represented, (6) the subject is dealt with comprehensively [38]. The responses were recorded with a four-point Likert scale ('very important,' 'rather important,' 'rather not important,' and 'not at all important').

iv. Satisfaction with the information related to COVID-19
    Information satisfaction was assessed by asking "How satisfied are you with the information you find on the Internet about the coronavirus?". Answers were recorded on a five-point Likert scale ('very dissatisfied,' 'dissatisfied,' 'partly,' 'satisfied,' and 'very satisfied').

**Internal consistency reliability of the scales.** The reliability statistics (Cronbach alpha) of the overall DHLI scale (15 items) is $\alpha = 0.88$ in this study. The alpha coefficient of the perception on the important factors in information searching (6 items) is $\alpha = 0.79$. These findings suggested an acceptable to good internal consistency of all the scales included in this study [39].

## Data analysis

Data were analyzed using SPSS (Version 25). Descriptive analysis was conducted for the socio-demographic characteristics, information search related variables, and the findings were reported with frequency, percentage, means, and standard deviation. The individual mean score of the five subscales of the DHLI was calculated. Subscale 5, "Protecting privacy", was excluded in the overall DHL mean score calculation because the items in this subscale are not applicable to all respondents [4]. Binary logistic regression analysis was carried out to identify the predictors of information satisfaction while controlling for age, gender, study program, study level, and financial support of the university students. Information satisfaction about COVID-19 (outcome variable) was grouped into two categories; high satisfaction (including 'satisfied' and 'very satisfied') and low satisfaction (including 'partly,' 'dissatisfied' and 'very dissatisfied'). The predictors were the mean score of the DHLI subscales, the importance of quality internet information search, sources of information utilized, and COVID-related topics searched. To check possible collinearity between the independent variables, the variance inflation factor (VIF) was assessed. Moreover, the association between DHL and the utilization of various information sources was analyzed using binary logistic regression. Adjusted odds ratios (AOR), 95% confidence intervals (95% CI), and $p$-values were reported. A $p$-value of $< 0.05$ was considered statistically significant.

## Results

A total of 5302 university students from China, Malaysia, and the Philippines responded to the survey. Table 1 presents the demographic characteristics and study information of the university students. The mean age of university students was 21.8 (SD = 4.01) years and approximately 75% were female. Among them, 52.6% were studying health science programmes. The majority of the students were studying at an undergraduate level (84.1%) and most of them were financially supported by their parents (85%).

The students' online information-seeking resources are reported in Table 2. Search engines (eg. Google, Bing, Yahoo) were highly utilized by the students (92%), which was followed by social media (88.4%). In contrast, websites of doctors or health insurance companies were less frequently used (64.7%). The current spread of the coronavirus (91.2%), symptoms of the disease (83.8%), and individual measures to protect against infection (70.1%) were the most frequently searched topics by university students. On the other hand, only 39.1% of university students reported searching for information on how to deal with psychological stress related to the COVID-19 pandemic. Approximately 61% of the university students reported that they were satisfied or very satisfied with the COVID-19 related information obtained.

The subscales and overall mean scores of the DHLI among 4913 university students who reported searching online for health information, is presented in the S1 Table. The overall mean score of the four DHL subscales was 2.89 (SD = 0.42), while overall scores of Chinese, the Philippines, and Malaysian students were 2.89 (SD = 0.43), 2.88 (SD = 0.43), and 2.93 (SD = 0.40) respectively. In the country specific analysis, the highest score was observed for the protecting privacy subscale across three countries (S1 Table).

**Table 1. Demographic variables of the university students from China, Malaysia, and the Philippines (n = 5302).**

| Variable | Frequency (n) | Percentage (%) |
|---|---|---|
| **Age (years)** *(Mean = 21.8; SD = 4.01)* | | |
| 18–25 | 4707 | 88.8 |
| 26–35 | 474 | 8.9 |
| ≥36 | 121 | 2.3 |
| **Gender**[*] | | |
| Male | 1288 | 24.5 |
| Female | 3973 | 75.5 |
| **Country** | | |
| China | 2042 | 38.5 |
| Malaysia | 953 | 18.0 |
| Philippines | 2307 | 43.5 |
| **Study Programme**[*] | | |
| Health sciences [a] | 2777 | 52.6 |
| Non-health sciences [b] | 2503 | 47.4 |
| **Study Level** | | |
| Undergraduate | 4461 | 84.1 |
| Post-graduate | 841 | 15.9 |
| **Sources of financial support** *(Multiple response)* | | |
| Support by parents | 4400 | 85.0 |
| Student grant | 833 | 16.1 |
| Employment during the semester | 722 | 14.0 |
| Employment during the semester break | 557 | 10.8 |
| Scholarship | 1450 | 28.0 |
| Other | 50 | 1.0 |

[*] Missing values: Gender (0.8%, n = 41); Study programme (0.4%, n = 22)

[a] Health-sciences including Medicine/Biomedical sciences, Allied Health Sciences (Nursing, Pharmacy, Medical Technology)

[b] Non-health sciences including Engineering Sciences/ICT, Linguistics & cultural studies/ communication arts, Mathematics/natural sciences, Law and economics/criminology/ public administration/political science, Social sciences/Social work/Psychology/Education, Business/Commerce/Accountancy/Management, Tourism/Hospitality/ Hotel management, Architecture/design/arts/visual studies, Secondary level/Others.

The university students' perception of the importance of several factors while searching for online health information is presented in the S2 Table. The overall mean score of importance is 3.62 (SD = 0.39). Verification of the information with regard to the Coronavirus and related topics was reported to be the most important factor when university students searched for the online information (M = 3.85, SD = 0.40). The presentation of different opinions on Coronavirus related information was ranked as least important by the respondents (M = 3.24, SD = 0.71). Predictor variables were tested for multicollinearity (all VIF < 10).

The association between the DHL subscales and the utilization of trustworthy information sources is reported in Table 3. Websites of public bodies, health portals (such as respective country's government health portals), websites of doctors or health insurance companies, news portals (such as newspapers and TV stations) were considered as trustworthy online information sources. Higher skills in adding self-generated content was positively associated with the utilization of trustworthy information sources about COVID-19; news portals among Chinese students (AOR 1.33, 95% CI 1.05–1.69), websites of doctors/ insurance companies

**Table 2. Online information seeking patterns among university students from China, Malaysia, and the Philippines (n = 4890).**

| Variable | Frequency (n) | Percentage (%) |
|---|---|---|
| **Sources used for online information seeking [a]** | | |
| **Search engines** | | |
| Low utilization | 394 | 8.0 |
| High utilization | 4505 | 92.0 |
| **Websites of public bodies** | | |
| Low utilization | 1554 | 31.8 |
| High utilization | 3330 | 68.2 |
| **Wikipedia and other online-encyclopaedias** | | |
| Low utilization | 2186 | 44.7 |
| High utilization | 2702 | 55.3 |
| **Social media** | | |
| Low utilization | 568 | 11.6 |
| High utilization | 4328 | 88.4 |
| **YouTube** | | |
| Low utilization | 1438 | 29.4 |
| High utilization | 3447 | 70.6 |
| **Blogs on health topics** | | |
| Low utilization | 2612 | 53.4 |
| High utilization | 2277 | 46.6 |
| **Online communities (eg. WhatsApp; Viber chat)** | | |
| Low utilization | 2985 | 61.2 |
| High utilization | 1894 | 38.8 |
| **Health portals** | | |
| Low utilization | 2568 | 52.6 |
| High utilization | 2317 | 47.4 |
| **Websites of doctors or health insurance companies** | | |
| Low utilization | 3160 | 64.7 |
| High utilization | 1721 | 35.3 |
| **News portals** | | |
| Low utilization | 881 | 18.0 |
| High utilization | 4001 | 82.0 |
| **COVID-related topics searched [a]** | | |
| Current spread of the coronavirus | 4460 | 91.2 |
| Symptoms of the disease COVID-19 | 4099 | 83.8 |
| Individual measures to protect against infection | 3429 | 70.1 |
| Transmission routes of the coronavirus | 3358 | 68.7 |
| Hygiene regulations | 3154 | 64.5 |
| Restrictions | 2725 | 55.7 |
| Current situation assessments and recommendations | 3061 | 62.6 |
| Economic and social consequences of the coronavirus | 2685 | 54.9 |
| Dealing with psychological stress caused by the coronavirus | 1910 | 39.1 |
| **Information satisfaction about COVID-19 [a]** | | |
| Very dissatisfied | 151 | 3.1 |
| Dissatisfied | 151 | 3.1 |
| Partly | 1587 | 32.5 |
| Satisfied | 2694 | 55.1 |

*(Continued)*

**Table 2.** (Continued)

| Variable | Frequency (n) | Percentage (%) |
|---|---|---|
| **Sources used for online information seeking** [a] | | |
| Very satisfied | 307 | 6.3 |

[a] Excluded sample who reported "No" in online health seeking information (7.3%, n = 389)

among Chinese (AOR 1.36, 95% CI 1.07–1.74) and Philippines students (AOR 1.68, 95%CI 1.39–2.02), and health portals among students from the Philippines (AOR 1.61, 95% CI 1.34–1.94) and Malaysia (AOR 1.47, 95% CI 1.02–2.12). Higher skills of evaluating reliability and determining relevance were positively associated with the utilization of trustworthy information sources especially for searching at the website of public bodies (Table 3).

Table 4 presents the factors associated with information satisfaction among university students. All models were adjusted for age, gender, country, study program, study level, and financial support of the university students. There was no collinearity among the variables.

Particularly, we tested the predictive relationships of DHL and the importance attached to web-based health information search towards information satisfaction. We also conducted cross-country analysis across the three sites (China, the Philippines, and Malaysia).

Country-specific analysis showed some similarities and differences in the predictors of information satisfaction among the three study sites (Table 4). For DHL, information searching (AOR = 1.83–1.98) and evaluating reliability (AOR = 1.41–3.23) were the most significantly associated with better information satisfaction. Protecting privacy in online information search was found to be positively related to information satisfaction among Chinese university students. The perceived importance of quality in internet information search remained a significant predictor of information satisfaction, except for respondents from Malaysia (Table 4).

Post-hoc power analysis was conducted using GPower version 3.1.9.4. Including all the study sites, regression analysis showed an OR of 1.86 for association between DHL information searching and information satisfaction, with a null prevalence of 0.40. With a total of 4,890 samples, a two-tailed test and a significance of 5%, the estimated power was 99.99%. This denotes that the probability of Type II error was reduced to 0.01.

## Discussion

In our study, we examined DHL, information-seeking behaviour, and its association with DHL as well as satisfaction with health information regarding COVID-19. With regard to the first research question, the study revealed that the overall mean score of DHL was 2.89. Similarly, findings from the Portuguese survey of the COVID-HL Network found that the DHL levels of university students in Portugal were above the third quartile of the scale [40]. A sufficient level of DHL in university students was also reported in the German survey of the COVID-HL Network, which assessed DHL among 14,916 from 130 universities [19]. Our finding was comparable with a study among Vietnamese university students, in that DHL mean score was 2.87 [41]. Although a different measurement scale was utilized in another study by Luo et al. (2018), the overall electronic health literacy among Taiwanese college students was reported to have medium and above [42].

Regarding the information-seeking behaviour, the majority of the students most often utilized search engines and social media. Search engines and social media were commonly used by university students in Egypt and Vietnam [41, 43]. Dramatic increment of searching the

**Table 3. Association between DHL and utilization of trustworthy information sources among university students from China, Malaysia, and the Philippines (n = 4890).**

| Digital Health Literacy Subscales | Websites of public bodies | | | | | | News portals | | | | | |
|---|---|---|---|---|---|---|---|---|---|---|---|---|
| | China | | Philippines | | Malaysia | | China | | Philippines | | Malaysia | |
| | AOR (95% CI) | p | AOR (95% CI) | P | AOR (95% CI) | P | AOR (95% CI) | P | AOR (95% CI) | p | AOR (95% CI) | p |
| Information searching | 0.90 (0.69–1.17) | 0.413 | 1.20 (0.90–1.62) | 0.216 | 0.76 (0.52–1.12) | 0.167 | 1.10 (0.81–1.49) | 0.542 | 0.85 (0.61–1.19) | 0.350 | 0.87 (0.53–1.43) | 0.593 |
| Adding self-generated content | 1.08 (0.87–1.33) | 0.497 | 1.23 (0.98–1.55) | 0.077 | 1.12 (0.80–1.57) | 0.504 | 1.33 (1.05–1.69) | 0.018 | 1.19 (0.91–1.55) | 0.204 | 1.11 (0.73–1.69) | 0.630 |
| Evaluating reliability | 1.46 (1.16–1.83) | 0.001 | 1.96 (1.52–2.53) | <0.001 | 1.57 (1.12–2.19) | 0.009 | 0.77 (0.59–1.02) | 0.066 | 1.10 (0.82–1.47) | 0.534 | 0.97 (0.63–1.50) | 0.886 |
| Determining relevance | 1.48 (1.10–1.99) | 0.010 | 0.92 (0.67–1.25) | 0.577 | 1.71 (1.10–2.67) | 0.018 | 2.30 (1.61–3.26) | <0.001 | 1.31 (0.92–1.86) | 0.136 | 1.36 (0.78–2.36) | 0.282 |
| Protecting privacy | 1.15 (1.00–1.31) | 0.046 | 1.11 (0.94–1.32) | 0.231 | 0.95 (0.74–1.21) | 0.655 | 1.37 (1.18–1.60) | <0.001 | 1.07 (0.87–1.31) | 0.514 | 1.19 (0.88–1.60) | 0.268 |

| Digital Health Literacy Subscales | Websites of doctors/insurance companies | | | | | | Health portals | | | | | |
|---|---|---|---|---|---|---|---|---|---|---|---|---|
| | China | | Philippines | | Malaysia | | China | | Philippines | | Malaysia | |
| | AOR (95% CI) | p | AOR (95% CI) | P | AOR (95% CI) | P | AOR (95% CI) | P | AOR (95% CI) | p | AOR (95% CI) | P |
| Information searching | 1.00 (0.74–1.34) | 0.984 | 0.78 (0.62–0.99) | 0.038 | 0.79 (0.53–1.16) | 0.224 | 1.07 (0.82–1.39) | 0.626 | 0.82 (0.65–1.03) | 0.082 | 0.73 (0.48–1.12) | 0.150 |
| Adding self-generated content | 1.36 (1.07–1.74) | 0.012 | 1.68 (1.39–2.02) | <0.001 | 0.92 (0.66–1.29) | 0.642 | 1.03 (0.83–1.28) | 0.772 | 1.61 (1.34–1.94) | <0.001 | 1.47 (1.02–2.12) | 0.039 |
| Evaluating reliability | 1.67 (1.29–1.16) | <0.001 | 0.98 (0.80–1.20) | 0.814 | 1.00 (0.72–1.40) | 0.988 | 1.62 (1.28–2.05) | <0.001 | 1.03 (0.84–1.26) | 0.778 | 1.24 (0.86–1.80) | 0.255 |
| Determining relevance | 1.15 (0.82–1.62) | 0.425 | 1.10 (0.86–1.40) | 0.457 | 1.41 (0.91–2.18) | 0.125 | 1.36 (1.00–1.85) | 0.052 | 0.99 (0.78–1.26) | 0.911 | 1.22 (0.76–1.97) | 0.414 |
| Protecting privacy | 0.77 (0.66–0.89) | 0.001 | 0.82 (0.72–0.95) | 0.006 | 0.89 (0.70–1.13) | 0.352 | 0.92 (0.80–1.05) | 0.207 | 0.92 (0.80–1.06) | 0.252 | 1.02 (0.78–1.33) | 0.884 |

topics about 'Coronavirus' or 'Pneumonia' on Google, Baidu Index, and Sina Weibo Index were observed in China at the beginning of the COVID-19 pandemic [44]. Similarly, the growing interest in the current COVID-19 pandemic and online information search about COVID-19 were reported in many countries in Asia, Europe, and the United States [16, 45–48]. The surveillance on trends in information seeking behaviour might be helpful to predict and tackle the spread and use of misinformation [16]. In the Philippines, university students reported that fear of getting infected is significantly associated with the utilization of Facebook as the information source ($p$ = 0.035) [49]. A mindful consideration should be taken to the usage of social media as it could amplify the misinformation and rumors especially during the COVID-19 pandemic [3, 50]. Our study findings highlighted that social media is commonly

**Table 4. Logistic regression of the predictors of information satisfaction among university students from China, the Philippines and Malaysia (n = 4890).**

| Variable | China | | Philippines | | Malaysia | |
|---|---|---|---|---|---|---|
| | Adj. OR (95% CI) | *p*-value | Adj. OR (95% CI) | *p*-value | Adj. OR (95% CI) | *p*-value |
| **Digital Health Literacy** | | | | | | |
| DHL: Information searching | 1.98 | <0.001 | 1.83 | <0.001 | 1.91 | 0.006 |
| | (1.49–2.63) | | (1.41–2.37) | | (1.21–3.01) | |
| DHL: Adding self-generated content | 1.11 | 0.357 | 0.99 | 0.896 | 1.49 | 0.038 |
| | (0.89–1.39) | | (0.81–1.21) | | (1.02–2.16) | |
| DHL: Determining relevance | 1.19 | 0.172 | 1.22 | 0.082 | 1.05 | 0.826 |
| | (0.93–1.52) | | (0.98–1.52) | | (0.71–1.54) | |
| DHL: Evaluating reliability | 2.50 | <0.001 | 1.41 | 0.013 | 3.23 | <0.001 |
| | (1.78–3.51) | | (1.08–1.84) | | (1.85–5.62) | |
| DHL: Protecting privacy in online search | 1.16 | 0.049 | 1.04 | 0.599 | 0.96 | 0.756 |
| | (1.00–1.35) | | (0.90–1.21) | | (0.72–1.27) | |
| **Importance of quality in internet information search** | 1.49 | 0.006 | 1.57 | 0.037 | 0.81 | 0.458 |
| | (1.12–1.98) | | (1.03–2.41) | | (0.47–1.41) | |

used among university students in China, the Philippines, and Malaysia. Hence, the students should be encouraged to utilize the reliable information sources such as government websites or university webpage.

Improving the DHL in population is one of the main pillars to combat the infodemic, which in turn could prevent the undesirable consequences of dis- and misinformation.

In our study, university students reported a high frequency of searching about the current spread of the coronavirus, symptoms, and individual measures to protect against COVID-19. COVID-19 pandemic was reported to have a psychological impact on the students [51–53]. However, the frequency of search for strategies to cope with psychological stress was low among university students. This topic was the uncommon with lowest frequency of search that could be found university students in Vietnam [41]. It might be explained by the time of this survey as it was conducted at an early phase of the pandemic. Mental health consequences became more prominent as the pandemic progressed [19]. Therefore, there is a need to consider improving the dissemination of information about possible consequences on mental health and resilience strategies to the university student population. Moreover, providing mental health awareness and support in the respective institution could be beneficial in the current pandemic [54].

With regard to the second research question, it could be found that the DHL skill of "adding self-generated content" was found to be positively associated with the frequent utilization of trustworthy online information sources among the university students in this study. As technology is used in various sectors in daily life, health literacy is extended as digital health literacy in the context of technology [55]. Our findings revealed the importance of developing skills to add user driven health content in web-based (social) media. Provision of training to the students to improve their ability to add self-generated content might be beneficial to access reliable information especially for Chinese students. The online platform and websites should provide space for self-generated content. Search engines might provide both reliable and unreliable information which should be aware of while recommending using the search engines. A recent study assessed the pattern of information dissemination about COVID 19 in search engines and reported that Google, Bing search engines mainly disseminate information from the legacy media, government, and healthcare information [56]. Therefore, mindful consideration should be taken in selecting search engines in adding self-generated content.

Evaluating online information with regard to its reliability, possible bias on commercial interest, and consistency in different websites are essential skills under the scope of DHL [4]. We found that determining relevance and evaluating reliability were positively associated with the utilization of trustworthy information sources about COVID-19. Our finding corresponds with prior research results. In their study, Chen et al. (2018) reported that lower health-literate individuals are more likely to use less-reliable sources, such as social media and blogs [57]. Similar findings were reported by Rosário et al. (2020) were Portuguese students with sufficient DHL skills in the area of reliability and validity assessment were more likely to search on public institution websites and health portals [40]. Among the German student population, sufficient skills in evaluating reliability was associated with online information searching at trustworthy sources including websites of public bodies [19].

Protecting personal identity and privacy is essential in the electronic health record and in joining online health-related groups, searching online information, and health-related applications [58]. Therefore, online (health) information seekers must understand the practices on how to ensure and protect one's own and other people's privacy [58]. Our findings revealed that safeguarding privacy was positively associated with the utilization of trustworthy online resources among Chinese and the Philippines students. Individuals with sufficient skills in protecting privacy could reduce the concern to share their personal information, health-related information, and increase participation in online health activities, and better evaluate the online resources' quality, credibility, and reliability [4, 58]. Our findings support this concept of privacy skills in DHL: The higher the private protection skills and the assessment of reliability, the more frequently trustworthy information sources are used.

Regarding the third research question, approximately 61% of the students reported that they were satisfied or very satisfied with the online information about COVID-19. During the COVID-19 pandemic, information satisfaction was reported approximately in half of the participants in Mainland China, Hong Kong, Macau, and Vietnam [35, 41]. Uncertainty of disease condition [59], rapid changes of recommendations [60], and amplification of information during the pandemic could lead to confusion, which in turn results in lower satisfaction.

DHL of university students especially in the area of information searching and evaluating reliability were associated with information satisfaction. Similarly, a study conducted among an elderly population reported that DHL was associated with information satisfaction [61]. Although the age and satisfaction level of the university students were different in our study and the elderly population study, DHL skills were found to be associated with the satisfaction with COVID-19 related information. Therefore, the association between of the different areas of DHL and information satisfaction is independent of age. Investigating DHL among the younger age group has positive effects on the future life course. Kor et al. found out that DHL was a significant predictor of information satisfaction in the adult population with chronic diseases [35]. Therefore, in the amidst of pandemic and infodemic, providing timely, relevant, and accurate information, promoting to utilize reliable information sources, and improving DHL to access, comprehend and appraise the online information is essential for the physical and psychological wellbeing.

## Conclusion and recommendations

Our study revealed that DHL is a key concept to utilize trustworthy online sources and to achieve higher satisfaction with online information sources, which in turn could be beneficial in making appropriate decisions related to their health. University students should be encouraged to utilize credible sources such as websites of public bodies, news portals (e.g., newspapers, national TV stations), websites of doctors/ insurance companies, and official health

portals for COVID-19 related information searching. Therefore, strengthening DHL among university students through e.g., online courses, trainings, and workshops, will enhance their critical thinking and help to empower them to use trustworthy information sources on COVID-19. Satisfaction with online information is important to have a better understanding of COVID-19 infection routes and to follow the regulations and precaution measures. Interventions should be designed to provide timely, reliable, and relevant COVID-19 related information targeting university students and their specific and genuine information needs. Awareness on mental wellbeing during the pandemic is limited. Therefore, online health talks or web-based information dissemination about the strategies for mental health challenges during the pandemic would be beneficial among the student population. The findings of this study could be integrated to the governments' strategies into disseminate COVID-19 related information to the public including university students.

### Strengths and limitations

To our knowledge, this is the first multi-countries study in East and South-East Asia to address the knowledge gap on the DHL and information searching behaviour among the university students during the COVID-19 pandemic. Valid and standardized data collection tools were used in this study. Sizable sample from three countries allows for more precise estimate of the findings. Furthermore, the estimated power was high (99.99%) in this study.

Our study had some limitations. Since our study was cross-sectional, we were not able to access the changes of DHL and information satisfaction over a period of time. The sample is self-selective based on a convenient sampling method. Hence, the generalization of the findings might be limited. We collected the data in the English language (except in China), which might limit the participation of students who are not fluent in English. Since the data were collected from university students, the findings might not be generalizable to people with lower educational attainment and other population groups (e.g., elderly).

### Supporting information

**S1 Table. Digital health literacy.**
(DOCX)

**S2 Table. Importance of internet information search.**
(DOCX)

**S1 File. Minimal underlying data set.**
(DOCX)

### Author Contributions

**Conceptualization:** Mila Nu Nu Htay, Ma. Carmen Tolabing, Kevin Dadaczynski, Orkan Okan, Angela Yee Man Leung, Tin Tin Su.

**Data curation:** Laurence Lloyd Parial.

**Formal analysis:** Laurence Lloyd Parial.

**Investigation:** Tin Tin Su.

**Methodology:** Kevin Dadaczynski.

**Project administration:** Ma. Carmen Tolabing, Kevin Dadaczynski, Orkan Okan, Angela Yee Man Leung, Tin Tin Su.

**Resources:** Ma. Carmen Tolabing.

**Software:** Laurence Lloyd Parial.

**Supervision:** Kevin Dadaczynski, Orkan Okan, Angela Yee Man Leung, Tin Tin Su.

**Writing – original draft:** Mila Nu Nu Htay.

**Writing – review & editing:** Mila Nu Nu Htay, Ma. Carmen Tolabing, Kevin Dadaczynski, Orkan Okan, Angela Yee Man Leung, Tin Tin Su.

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
