## [Decision Letter · Decision Letter 0]

16 Nov 2021

PONE-D-21-32546Digital health literacy, online information-seeking behaviour, and  satisfaction of  Covid-19 information among the university students of East and South-East AsiaPLOS ONE

Dear Dr. Su,

Thank you for submitting your manuscript to PLOS ONE. After careful consideration, we feel that it has merit but does not fully meet PLOS ONE’s publication criteria as it currently stands. Therefore, we invite you to submit a revised version of the manuscript that addresses the points raised during the review process.

We look forward to receiving your revised manuscript.

Kind regards,

Bijaya Kumar Padhi, PhD, MPH

Academic Editor

PLOS ONE

Reviewers' comments:

Reviewer's Responses to Questions

**Comments to the Author**

1. Is the manuscript technically sound, and do the data support the conclusions?

Reviewer #1: Yes

Reviewer #2: Partly

2. Has the statistical analysis been performed appropriately and rigorously? 

Reviewer #1: Yes

Reviewer #2: No

3. Have the authors made all data underlying the findings in their manuscript fully available?

Reviewer #1: Yes

Reviewer #2: No

4. Is the manuscript presented in an intelligible fashion and written in standard English?

Reviewer #1: Yes

Reviewer #2: Yes

5. Review Comments to the Author

Reviewer #1: A relevant study carried out during Covid19 pandemic. Introduction and methodology part is written extensively. Rigorous Data Analysis carried out. Minor grammatical errors need to be corrected. Conclusion and recommendations still need to be short & concise.

Reviewer #2: The authors have assessed the digital heatlh literacy, its association with trustworthy online information source utilization, and predictors of information satisfaction among university students in Malaysia, China and The Phillipines. The paper has some commendable strengths - it represents information from a sizeable sample size. It also uses a standardized instrument for collecting information (not sure if this has been used/ tested for validity in the countries studied, though). The study has some interesting findings to report.

However, the paper does not provide a convincing rationale for doing the study. It does not rationalize the selection of context and participant profile (except that 'information is scarce' from the region(s)), and why participants from the three countries could/ should be 'studied together'. Is there a reason to expect that these countries would have unique findings and if yes, why so?

The paper does not specify how the variables were chosen for the initial data collection and further, for model fitting (pragmatically or after an initial bivariate analysis). Consequently, it remains unclear if it could have made sense to build separate country-specific models with a wider set of variables, some of which could have been unique to the country context?

The discussion, recommendations and conclusion do not adequately highlight what new knowledge this study adds to the existing pool of knowledge.

The broader issue of concern is that 'satisfaction' is a very subjective assessment - a well referenced sentence (or two) on how local and respective university culture, regulations and COVID-19 epidemiological situation in the country could have impacted information satisfaction, would have helped the reader to reflect better on the findings and recommendations.

In view of the above, the paper requires significant revision (reanalysis and rewriting) and cannot be accepted in its current shape.

6. PLOS authors have the option to publish the peer review history of their article (what does this mean?). If published, this will include your full peer review and any attached files.

Reviewer #1: No

Reviewer #2: **Yes: **Archisman Mohapatra

---

## [Author Response · Author response to Decision Letter 0]

9 Jan 2022

We have prepared the manuscript and file names according to PLOS ONE’s requirement. 

2. Upon re-submitting your revised manuscript, please upload your study’s minimal underlying data set as either Supporting Information files or to a stable, public repository and include the relevant URLs, DOIs, or accession numbers within your revised cover letter. For a list of acceptable repositories. 

We have included in this resubmission the minimal underlying data set as Supporting Information files. 

We revised to ensure that ethic statement is only in the Methods section. 

4. Please include captions for your Supporting Information files at the end of your manuscript, and update any in-text citations to match accordingly.

We have included captions for Supporting Information files at the end of the manuscript. 

Frist of all we would like to thanks to the reviewers for your valuable time and providing insightful comments for improvement. We have addressed all the comments and provided explanation for each point. 

Reviewer #1: 

1. A relevant study carried out during Covid19 pandemic. Introduction and methodology part is written extensively. Rigorous Data Analysis carried out. Minor grammatical errors need to be corrected. Conclusion and recommendations still need to be short & concise.

Thank you very much for the suggestions. We revised the conclusion and recommendation to be short and concise. (page 25 in the manuscript with tract changes).

Reviewer #2: 

1. The authors have assessed the digital heatlh literacy, its association with trustworthy online information source utilization, and predictors of information satisfaction among university students in Malaysia, China and The Phillipines. The paper has some commendable strengths - it represents information from a sizeable sample size. It also uses a standardized instrument for collecting information (not sure if this has been used/ tested for validity in the countries studied, though). The study has some interesting findings to report.

However, the paper does not provide a convincing rationale for doing the study. It does not rationalize the selection of context and participant profile (except that 'information is scarce' from the region(s)), and why participants from the three countries could/ should be 'studied together'. Is there a reason to expect that these countries would have unique findings and if yes, why so?

The previous studies reported that health literacy level is limited in these countries compared to the other neighbouring countries. Previous studies in Southeast Asia reported a high prevalence of limited health literacy in Malaysia (94.2% among caregivers in 2015 and 85.8% among type 2 DM patients in 2016), while in the neighbour countries such as Singapore (43.7%), Thailand (48.7%) and Myanmar (28.2%) a lower percentage could be found among patients, caregivers, and general population (21-25). A study conducted among middle school students in the Philippines before the COVID-19 pandemic reported limited health literacy especially on personal health and on prevention and control of diseases (26). A meta-analysis of health literacy among Chinese students revealed that only 26% had an adequate health literacy level (27). The wealth of information provided during the COVID-19 pandemic caused confusion and negative impacted individual prevention and control measures especially in limited health literate population groups. Therefore, there is a need to investigate DHL and information seeking pattern during the pandemic period in Asian countries. We included the justification for the selection of study context in the introduction section. (Page 5 in the manuscript with tract changes)

References

1. Chan HK, Hassali MA, Lim CJ, Saleem FJJoPHSR. Exploring health literacy and difficulty in comprehending pediatric medication labels among caregivers in Malaysia: a pilot study. 2015;6(3):165-8.

2. Azreena E, Suriani I, Juni MH, Fuziah P. Factors associated with health literacy among Type 2 Diabetes Mellitus patients attending a government health clinic, 2016. International Journal of Public Health and Clinical Sciences. 2016;3(6).

3. Zhang XH, Li SC, Fong KY, Thumboo J. The impact of health literacy on health-related quality of life (HRQoL) and utility assessment among patients with rheumatic diseases. Value Health. 2009;12 Suppl 3:S106-9.

4. Wannasirikul P, Termsirikulchai L, Sujirarat D, Benjakul S, Tanasugarn C. HEALTH LITERACY, MEDICATION ADHERENCE, AND BLOOD PRESSURE LEVEL AMONG HYPERTENSIVE OLDER ADULTS TREATED AT PRIMARY HEALTH CARE CENTERS. Southeast Asian J Trop Med Public Health. 2016;47(1):109-20.

5. Oo WM, Soe PP, Lwin KT. Status and determinants of health literacy: a study among adult population in selected areas of Myanmar. International Journal of Community Medicine and Public Health. 2015;2(3).

6. Javier R, Tiongco M, Jabar M. How Health Literate are the iGeneration Filipinos? Health Literacy Among Filipino Early Adolescents in Middle Schools. Asia-Pacific Social Science Review. 2019;19(3).

7. Mao Y, Xie T, Zhang N. Chinese Students' Health Literacy Level and Its Associated Factors: A Meta-Analysis. Int J Environ Res Public Health. 2020;18(1).

2. The paper does not specify how the variables were chosen for the initial data collection and further, for model fitting (pragmatically or after an initial bivariate analysis). Consequently, it remains unclear if it could have made sense to build separate country-specific models with a wider set of variables, some of which could have been unique to the country context?

We revised the Measures section to explain the initial data gathered for this study: 

The study utilized the COVID-HL University Students Questionnaire, which was developed by the COVID-Health Literacy Consortium 1 and then translated and adapted in the context of this study. The questionnaire is composed of several sections pertaining to sociodemographic information, life situation and future anxiety, health literacy and information seeking behavior, and personal health situation. Following the study objectives, we gathered the participants’ sociodemographic variables, online information-seeking behaviour, DHL, and satisfaction with the information related to COVID-19. (Page 9 on manuscript with tract changes).

We included the following statements to discuss the variables chosen for the combined analysis. We also conducted a country-specific analysis including other potential predictors of information satisfaction. (Page 17-20 in the manuscript with tract changes)

Table 4 presents the factors associated with information satisfaction among university students. All models were adjusted for age, gender, country, study program, study level, and financial support of the university students. There was no collinearity among the variables. 

Particularly, we tested the predictive relationships of DHL and importance of quality internet information search toward information satisfaction. We also conducted cross-country analysis among the three sites (China, the Philippines, and Malaysia). (Page 17-20 in the manuscript with tract changes)

Three subscales of DHL were significantly associated with the information satisfaction among the university students. An increase in skill unit resulted in a higher probability of information satisfaction between AOR = 1.19 to 1.86 with the highest AORs found for information search and evaluating reliability (Table 4). 

Meanwhile, country-specific analysis showed some similarities and differences in the predictors of information satisfaction among the three study sites (Table 4). For DHL, information searching (AOR = 1.83 – 1.98) and evaluating reliability (AOR = 1.41 – 3.23) were the most significantly associated with better information satisfaction. Meanwhile, protecting privacy in online search was found to be positively related to information satisfaction among the Chinese university students. The perceived importance of quality information in online search remain a significant predictor of information satisfaction, except for the participants from Malaysia. (Page 19 in the manuscript with tract changes). 

3. The discussion, recommendations and conclusion do not adequately highlight what new knowledge this study adds to the existing pool of knowledge.

We revised the discussion highlighting the new knowledge revealed in our study. The recommendation and conclusion have been revised. (Page 21-25 in the manuscript with tract changes).

4. The broader issue of concern is that 'satisfaction' is a very subjective assessment - a well referenced sentence (or two) on how local and respective university culture, regulations and COVID-19 epidemiological situation in the country could have impacted information satisfaction, would have helped the reader to reflect better on the findings and recommendations.

We included the COVID-19 epidemiological situation in the study countries, regulations, disease knowledge level among the study population, and their culture and behaviour during COVID-19 pandemic. (Page 6-7 in the manuscript with tract changes)

---

## [Editor Report · Decision Letter 1]

18 Mar 2022

Digital health literacy, online information-seeking behaviour, and  satisfaction of  Covid-19 information among the university students of East and South-East Asia

PONE-D-21-32546R1

Dear Dr. Su,

We’re pleased to inform you that your manuscript has been judged scientifically suitable for publication and will be formally accepted for publication once it meets all outstanding technical requirements.

Kind regards,

Bijaya Kumar Padhi, PhD, MPH

Academic Editor

PLOS ONE
---

## [Editor Report · Acceptance letter]

25 Mar 2022

PONE-D-21-32546R1 

Digital health literacy, online information-seeking behaviour, and satisfaction of Covid-19 information among the university students of East and South-East Asia 

Dear Dr. Su:

I'm pleased to inform you that your manuscript has been deemed suitable for publication in PLOS ONE. Congratulations! Your manuscript is now with our production department. 

Kind regards, 

on behalf of

Dr. Bijaya Kumar Padhi 

Academic Editor

PLOS ONE